# Hyperpolarization of Nitrile Compounds Using Signal Amplification by Reversible Exchange

**DOI:** 10.3390/molecules25153347

**Published:** 2020-07-23

**Authors:** Sarah Kim, Sein Min, Heelim Chae, Hye Jin Jeong, Sung Keon Namgoong, Sangwon Oh, Keunhong Jeong

**Affiliations:** 1Department of Chemistry, Seoul Women’s University, Seoul 01797, Korea; srh0714@daum.net (S.K.); sein5762@naver.com (S.M.); mek2425@naver.com (H.C.); sknam@swu.ac.kr (S.K.N.); 2Department of Chemistry, Korea Military Academy, Seoul 01805, Korea; hyejinj1011@naver.com; 3Korea Research Institute of Standards and Science, Daejeon 34113, Korea

**Keywords:** parahydrogen, SABRE, nitrile, hyperpolarization

## Abstract

Signal Amplification by Reversible Exchange (SABRE), a hyperpolarization technique, has been harnessed as a powerful tool to achieve useful hyperpolarized materials by polarization transfer from parahydrogen. In this study, we systemically applied SABRE to a series of nitrile compounds, which have been rarely investigated. By performing SABRE in various magnetic fields and concentrations on nitrile compounds, we unveiled its hyperpolarization properties to maximize the spin polarization and its transfer to the next spins. Through this sequential study, we obtained a ~130-fold enhancement for several nitrile compounds, which is the highest number ever reported for the nitrile compounds. Our study revealed that the spin polarization on hydrogens decreases with longer distances from the nitrile group, and its maximum polarization is found to be approximately 70 G with 5 μL of substrates in all structures. Interestingly, more branched structures in the ligand showed less effective polarization transfer mechanisms than the structural isomers of butyronitrile and isobutyronitrile. These first systematic SABRE studies on a series of nitrile compounds will provide new opportunities for further research on the hyperpolarization of various useful nitrile materials.

## 1. Introduction

Nuclear Magnetic Resonance (NMR) is a powerful tool for identifying the (in)organic and biological structures of molecules in various states. The study of the dynamics of molecules is a key subject in the area of NMR studies with specific pulse sequences. One of the prominent applications of this NMR method is Magnetic Resonance Imaging (MRI), which is a precise, non-invasive body-scanning technique. The signals in NMR depend on the population differences in magnetic-field-induced Zeeman splitting, which is proportional to the external field, and follows a Boltzmann distribution. Hence, a strong magnetic field is needed. This results in high costs in purchasing a strong magnet as well as maintaining the superconducting magnet. To overcome these shortcomings, hyperpolarization (HP) techniques have been developed to increase the signal strength [1,2,3,4]. Along with other well-known hyperpolarization techniques, parahydrogen-induced hyperpolarization, which uses a higher proportion of parahydrogen gas than orthohydrogen, has been proposed as a potential cost-efficient and rapid technique, with processing in non-harsh conditions. Hyperpolarization can be achieved using spin-state energy transfer from polarized antiparallel spins of parahydrogen [5,6]. Its benefits are particularly significant in focusing on various molecular structures and medical applications such as MRI. Parahydrogen-induced polarization (PHIP) uses the hydrogenation reaction on an unsaturated bond, which induces hyperpolarization through additional covalent bonding on carbon [7,8,9]. This process results in structural changes after the hydrogenation reaction. While signal amplification by reversible exchange (SABRE), which is another way of using parahydrogen, employs a non-hydrogenation parahydrogen-based hyperpolarization technique [10,11,12], SABRE relies on the temporary bonding of the substrate at the metal center (Iridium) to enable hyperpolarization creation on the nucleus spins of the substrates. With these powerful characteristics, most studies using SABRE have focused on specific structures with sp^2^ nitrogen atoms, such as pyridine, purines, diazirines, triazole, or their derivatives [13,14,15,16,17,18]. Previous studies helped in setting up the polarization mechanism, and its trend in the polarization for the structures has been studied over the last few decades [19,20,21,22,23]. On the other hand, newly discovered structures lack these perspectives. Hence, reports on detecting small concentrated materials and reaction mechanisms that have the potential for application in the near future are important [24,25,26,27]. In this context, polarization with nitrile is quite new and studies focusing on nitrile compounds have not been conducted systemically, other than for the aromatic and nonaromatic nitrile materials with low enhancement factors [28,29]. Furthermore, ^13^N and ^13^C polarization transfer via nitrile groups (short range coupling) has recently attracted interest due to its potentially long relaxation time (T1) [30,31]. Therefore, its systematic study with nitrile groups is expected to render more insights into the polarization transfer mechanism, such as SABRE-SHEATH (SABRE in SHield Enables Alignment Transfer to Heteronuclei), for a heteronuclear system in the future. The nitrile is an organic compound that has a −C≡N functional group in the structure. Nitriles are found in many useful organics including nitrile gloves, nitrile-containing polymers used in laboratory work, and methyl cyanoacrylate, which is used in super glue. More importantly, over 30 nitrile-containing pharmaceuticals are now prescribed all over the world [32,33]. The core structure, i.e., the nitrile group, is robust, and it is not well metabolized. It can stay for a long time in the body [34,35]. Overall, this material is widely used not only in industry but also in fundamental research and needs to be investigated more systematically and extensively.

Here, we present a systematic hyperpolarization study on nitriles using SABRE to obtain the maximum hyperpolarization in different magnetic fields and concentrations. Our study unveiled some unprecedented results, which showed much more signal enhancement than that ever reported before and polarization trends with different carbon attachments.

## 2. Results and Discussion

### 2.1. SABRE on Acetonitrile

After the activation of an Ir-catalyst (3.12 µmol) in bubbling parahydrogen for 20 min, different volumes of acetonitrile were added to the solution (MeOH-*d*_4_ 900 µL). The polarization transfer from parahydrogen to acetonitrile was performed in the solenoid coil, which was matched with different magnetic fields. Subsequent experiments with other nitrile compounds were performed using the same procedure (Figure 1).

The optimization for enhancing the polarization-transfer efficiency of acetonitrile was carried out with different magnetic fields with different concentrations of acetonitrile in MeOH-*d*_4_ solution. According to the results (Figure 2), all three protons in the methyl group on acetonitrile are hyperpolarized and its polarization is maximized to approximately 70 G in the solution. The signal enhancement ratio is calculated by comparing the thermal integral with the hyperpolarized integral obtained by SABRE (see the Appendix A).

It is worth noting that the enhancement factor of more than 130-fold in the hyperpolarization is much bigger than any previously reported result. For example, it is only 8-fold in the case of acetonitrile. However, it can be enhanced as much as 60 times by adding pyridine-*d*_5_ (20 times for pyridine), which was introduced as a co-substrate. Further studies on this co-substrate with more nitrile compounds would be needed in the future [28]. Furthermore, its polarization is dependent on the concentration and is maximized with 5 μL of acetonitrile in 900 μL of solvent, which is normally attributed to the chemical exchange dynamics with optimized molar concentrations [36]. This is the main reason behind the polarization enhancement, and its efficiency is also related to the relaxation effect induced by the Ir-catalyst including the proton T_1_ time. Its phase trend (Figure 3) is changed from the earth magnetic field (~0.5 G) and reversed again by increasing the magnetic field. This is different from the previously reported phenomenon [28] and seems to be slightly different in terms of the polarization transfer mechanism from parahydrogen and needs further investigation. It is speculated to be because of the different matching condition of the magnetic field with different *J–J* coupling from both hydrides in the Ir-catalyst. We found that the concentration showing the maximum polarization number at the same magnetic field was different. In the earth field, a small amount of acetonitrile showed the biggest polarization, whereas there was a trend of a higher concentration of substrate leading to larger polarization in the higher magnetic field. This indicates that the polarization efficiency depends not only on the magnetic field but also on the concentration of the substrate in acetonitrile SABRE.

### 2.2. SABRE on Propionitrile

The different aspect of propionitrile compared to acetonitrile is that one more methyl group is attached than in acetonitrile, which can be polarized by the polarization transfer from the protons from H-1. Therefore, it causes less polarization on the H-2 than the H-1, which is supported by our result (Figure 4). The maximum enhancement factor is approximately 130-fold, which is shown in the 70 G in 1 μL and 120 G in 3 μL of propionitrile, which has almost the same trend as acetonitrile (the concentration for each volume is indicated in Appendix A). However, more interestingly, the polarization in the earth field is bigger than 20 G in all the H-2 protons, whereas the H-1 showed the same trend as the acetonitrile case. This can be attributed to the polarization transfer through the long range *J–J* coupling [20], which can be matched with a low magnetic field. This is supported by the smaller polarization enhancement than of the protons in H-1. However, the protons in H-1 can be hyperpolarized in higher magnetic fields, and its polarization is transferred into the H-2. Even though the polarization transfer is active from H-1 to H-2, the enhancement factor of H-1 in propionitrile is similar to that of acetonitrile.

### 2.3. SABRE on Butyronitrile and Isobutyronitrile

These two structures are structural isomers with differently branched structures. The largest enhanced signals from protons were on H-1 in both structures—with an approximately 100-fold enhancement, which showed almost the same trend with propionitrile and acetonitrile. In both cases, 5 μL was the optimal volume for obtaining the highest polarization (Figure 5). For the polarization-transfer efficiency, polarization is transferred to the other protons from H-1 in butyronitrile. However, polarization transfer in isobutyronitrile is much less compared to that in butyronitrile case. This indicates that the polarization efficiency is closely related to the polarization-transferring proton number (two protons in butyronitrile vs. one proton in isobutyronitrile) and adjacent numbers of protons (polarization-transferred protons) that are coupled (two protons in butyronitrile vs. six protons in isobutyronitrile). This reveals that a linear type of nitrile or other functional groups that are chelated with the Ir-catalyst would be a better option for achieving higher polarization-transfer efficiency.

### 2.4. SABRE on Valeronitrile

We additionally performed SABRE on valeronitrile, which contains one more methyl group in the nitrile than butyronitrile. Its optimal polarization factor with different magnetic fields and concentrations also showed the same trend as the butyronitrile case—an approximately 100-fold enhancement in 70 G with a 5 μL amount of valeronitrile. It is hard to discern two hyperpolarized signals from the methyl group of H-2 in the spectrum, which have a small chemical shift difference. Therefore, we calculated the enhancement factor on all five hydrogens from H-2 of valeronitrile’s structure. Almost all the trends were similar to those for butyronitrile, except strong hyperpolarization on all protons in the 120 G for 1 μL of substrate, which may be attributed to the change in the chelating strength caused by the increased weight and coupling strength. (Figure 6) However, this needs to be studied further.

## 3. Materials and Methods

60 MHz ^1^H spectra were obtained with Spinsolve Ultra (Magritek, Abingdon, UK). This device was automated with a 5% D_2_O sample in shimming, and the NMR signal was checked using a Wilkinson’s catalyst. The parahydrogen generator was activated by a home-built instrument [37]. Hydrogen gases (a spin isomer mixture of ortho-H_2_ and para-H_2_) were passed through heat exchangers filled with Fe(OH)O catalysts in a liquid nitrogen dewar to obtain up to 50% p-H_2_.

An Ir-catalyst, (Ir(COD)(IMes)Cl) (COD=η^4^-1,5-cyclooctadiene and IMes=1,3-bis(2,4,6-trimethylphenyl)imidazole-2-ylidene) [16,17] and nitrile compound for each concentration (1, 3, 5, and 7 μL of acetonitrile, corresponding to 19, 57, 96, and 134 μmol, respectively; all concentration amounts are tabulated in Appendix A) were added to the NMR tube. After dissolving in 900 μL of deuterated methanol, the solution was bubbled through the tube with para-hydrogen for 20 min [38]. The mixture was foamed for 1 min at each magnetic field before being moved to the NMR spectrometer. The spectrum was obtained at 60 MHz. NMR data were processed and analyzed using the Mnova software (12.0.2, Masterlab Research, S.L, Abingdon, UK).

## 4. Conclusions

In this study, we applied the promising hyperpolarization technique SABRE on a series of nitrile compounds with an additional methyl group attached. By performing SABRE with various magnetic fields and substrate concentrations, we unveiled its hyperpolarization characteristics and properties to maximize the polarization and its transfer efficiency. Through this sequential study, we could conclude that the polarization on hydrogen decreases with increasing distance from the nitrile group and its highest polarization is observed at approximately 70 G and with 5 μL of substrates in all structures. Furthermore, we obtained the best enhancement factor of more than 130-fold for various nitrile compounds using 50 percent of parahydrogen. Therefore, we can expect enhanced polarization numbers in the future using a higher concentration of parahydrogen. This study has only been performed in methanol, which is not a biological solvent. To be applicable in biological systems, a hyperpolarization study in biological solvent systems or new strategy for obtaining hyperpolarization in an aqueous solvent would be needed [39,40,41]. Interestingly, the more branched structures in the ligand showed less effective polarization-transfer mechanisms than the butyronitrile and isobutyronitrile SABRE results. Furthermore, upon investigating several cases, the H-2 protons from nitrile compounds showed higher polarization, which is contrary to previously reported cases. It is noteworthy that the effective polarization transfer may be possible using low or ultralow magnetic fields, which are normally of approximately 100 G. These first-ever systematic SABRE studies on a series of nitrile compounds will certainly widen the new possibilities for more hyperpolarizable chemical materials using SABRE in the future.

## Figures and Tables

**Figure 1 molecules-25-03347-f001:**
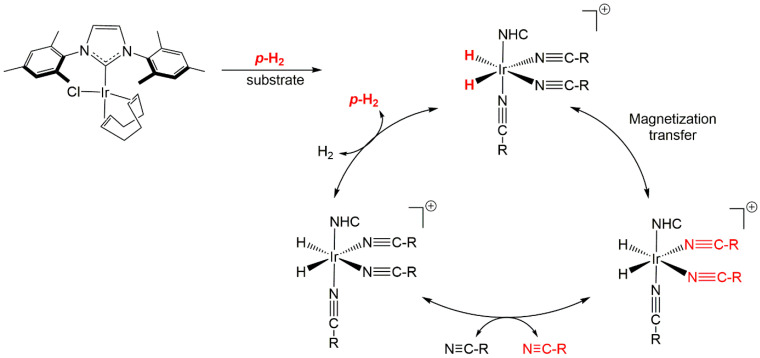
Hyperpolarization procedure of Signal Amplification by Reversible Exchange (SABRE) with nitrile compounds [12]. NHC is *N*-heterocyclic compounds.

**Figure 2 molecules-25-03347-f002:**
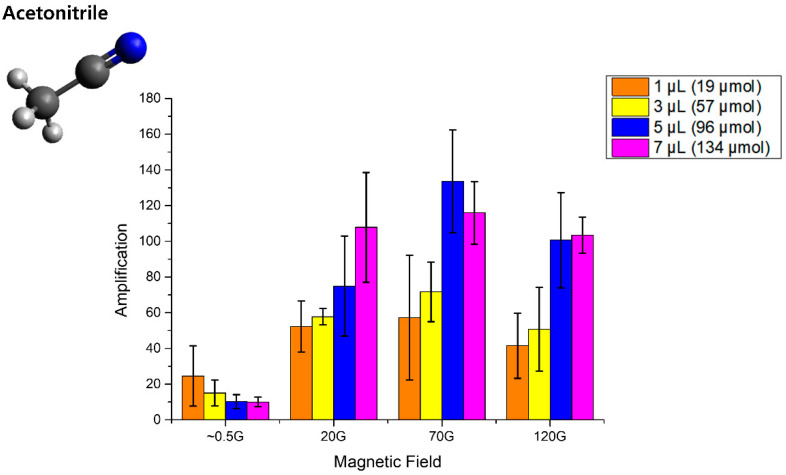
Amplification factors according to different magnetic fields for different amounts of acetonitrile in the solvent (MeOH-*d*_4_).

**Figure 3 molecules-25-03347-f003:**
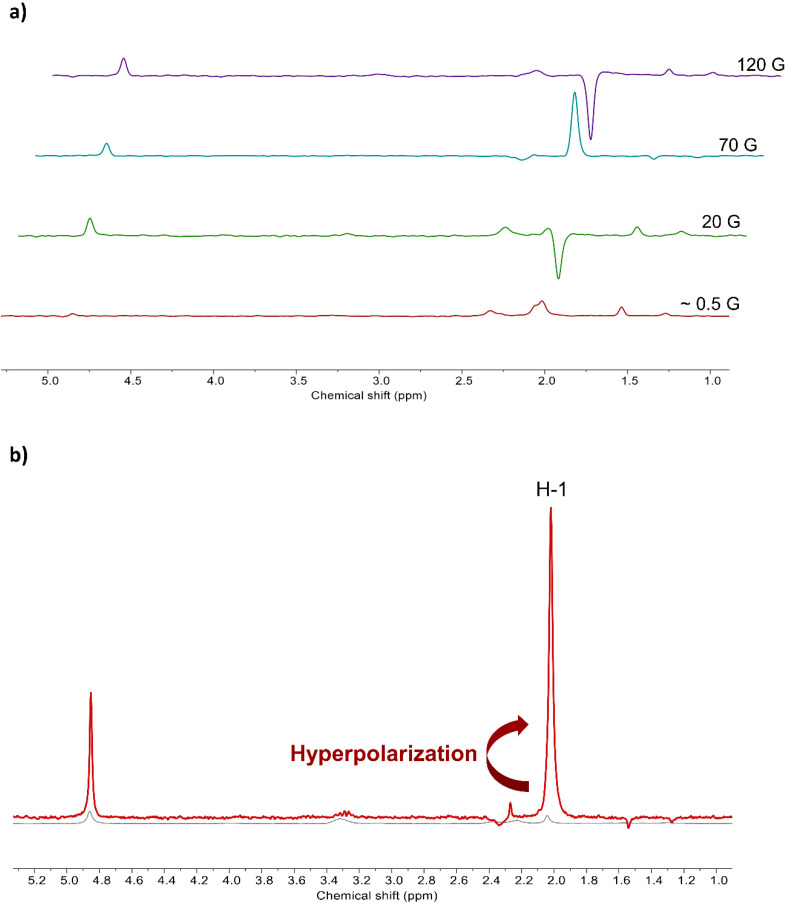
NMR spectrum after SABRE in different magnetic fields on the 1 μL (19 μmol) of acetonitrile (**a**) and hyperpolarized spectrum in 70 G (**b**) All other examples of spectra are shown in Appendix A.

**Figure 4 molecules-25-03347-f004:**
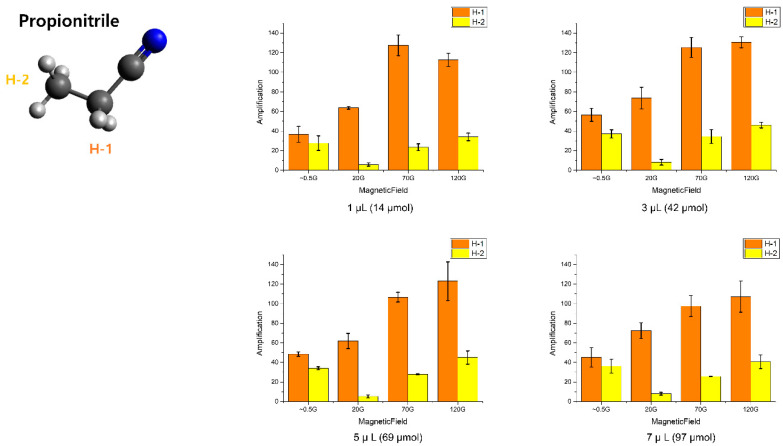
Amplification factors according to the different magnetic fields for different amounts of propionitrile in solvent (MeOH-*d*_4_).

**Figure 5 molecules-25-03347-f005:**
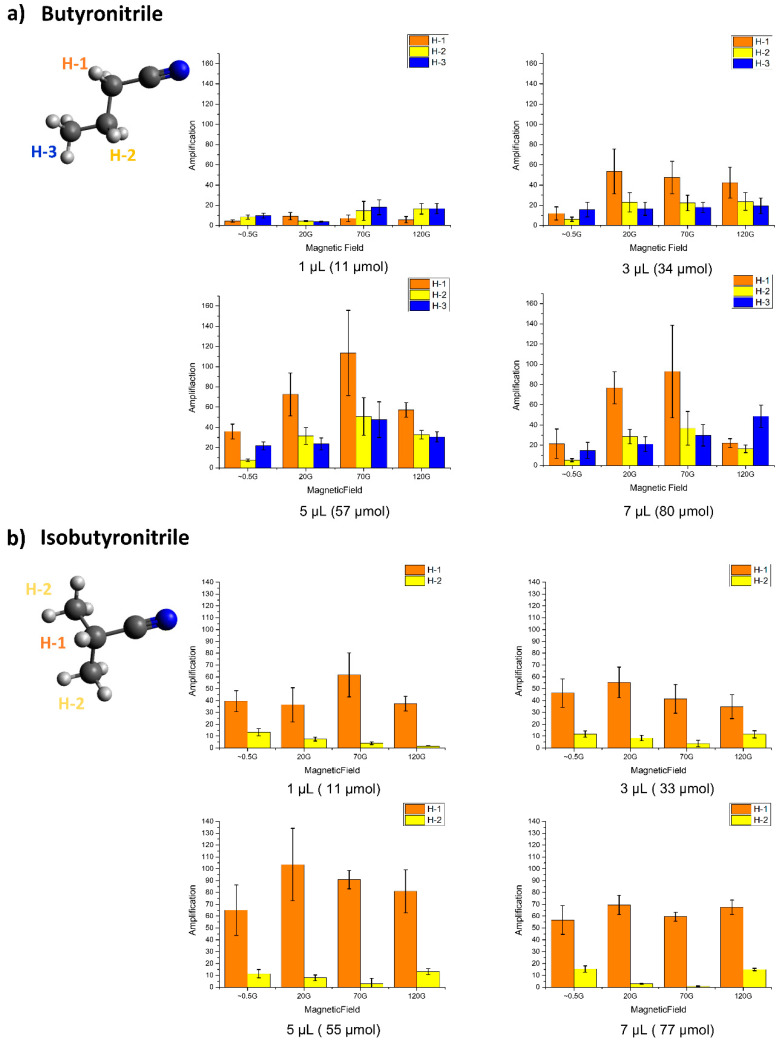
Amplification factors according to the different magnetic fields for different amounts of (**a**) butyronitrile and (**b**) isobutyronitrile in solvent (MeOH-*d*_4_).

**Figure 6 molecules-25-03347-f006:**
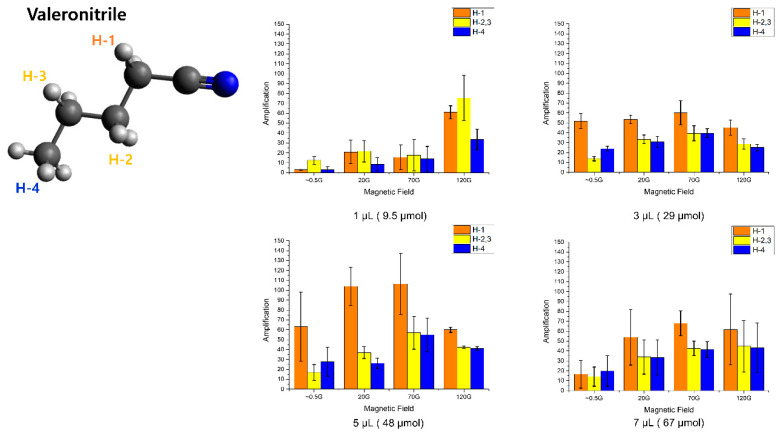
Amplification factors according to the different magnetic fields for the different amounts of valeronitrile in solvent (MeOH-*d*_4_).

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
