# Peer review of "Hyperpolarization of Nitrile Compounds Using Signal Amplification by Reversible Exchange"

_molecules, 2020, doi:10.3390/molecules25153347_

Round 1

Reviewer 1 Report

The authors present an explorative and systematic study of the 1H SABRE hyperpolarization of a series of nitrile-containing compounds, which is of merit. While the results are interesting, significant revisions to the presentation of the data need to be addressed to make the manuscript accessible to the reader. In general, there are grammatical errors that should be addressed, although these are minor. Questions and comments are listed below by line number.

Questions and comments:

[34/35]:  A brief explanation of why larger magnetic fields generate larger population differences should be included.

[38/39]: Describe how parahydrogen is used as a source of spin polarization.

[50]: This sentence needs references, and it is best to give an idea of how “long” (eg. Over the last decade).

[52/53]: Both detection of small concentrated materials and reaction mechanism studies have been shown with SABRE, consider adding references to this statement.

[16,53-55]: While a systematic study of the 1H hyperpolarization of nitriles is new, nitrile hyperpolarization by SABRE has been extensively investigated in the heteronuclear SABRE community, so the use of nitriles is not necessarily “new”. What is unique is that this is a very different coupling regime (long range 5JHH) to generate hyperpolarization than in heteronuclear SABRE (for instance 2JNH or 3JCH in nitriles). This should be considered when formulating the arguments in the introduction.

[Entire manuscript]: All instances of volumes of hyperpolarization targets should be replaced by molar concentration, as should the concentration of the iridium catalyst.

[Fig. 1]: Figure 1 is too similar to the Figure 1 in reference [12].

[Fig. 2]: The category axis (1,2,3,4) needs a label or the substrate concentrations should be used as the axis increments. Also, what is “normal”?

[Fig. 2]: How are your “amplification” factors calculated? Are they compared to thermal spectra at a given substrate concentration?

[85]: Reference [19] demonstrates that the addition of pyridine greatly increases the enhancement up to 60 for pyridine-d5. This should be acknowledged here.

[87]: “Furthermore polarization is dependent on the concentration… which is normally attributed to the chemical exchange difference”. What do you mean by this? References are needed here.

[89/90]: “phase trend (Figure 3) is reversed… by increasing the magnetic field”. This is not the case for the spectrum at 70G in Figure 3 and ref. [19] shows that this is not the case for acetonitrile. How does this match the “traditional mechanistic explanation”?

[92-94]: “phase change is also seen with different concentrations of the substrate for different magnetic fields… may stem from the two different spin states of hydrogen”. This could only happen if you had enriched orthohydrogen, which is not generated by thermal cooling. Why then should the phase change as a function of the substrate concentration?

[95-98]: The data you show in Figure 2 disagrees with this statement… it looks like you achieve maximum polarization at 70G independent of concentration. An estimate on the error in these measurements should be made.

[Fig. 3, top]: The frequency axis and magnetic field labels are nearly impossible to read. Consider removing the grid from the spectrum (it is not needed here) and perhaps using a larger stack angle so that the spectra can be zoomed in the vertical direction. Also, you should make sure that the same vertical zoom is used for each of the spectra (the 0G spectrum is clearly not on the same scale, which can be seen by the noise). Also, the label should be ~0.5G, not 0G.

[Fig. 3, bottom]: The spectra are either not on the same scale, which falsely generates a “hyperpolarized” spectrum, or line broadening was not applied equivalently to both spectra, which will change the apparent signal to noise. Example spectra should be of acetonitrile, not valeronitrile which does not come until the end of the paper.

[Fig. 4/5/6]: The data should only be shown in one format, it is confusing otherwise. While the 3D bar charts are look nice, the bars hide the data behind them and it is hard to actually see the trends in the data. Instead, 2D bar charts separated by substrate concentration would be a much more appropriate visualization of the data. Red/green combinations should be avoided in figures to aid people with red/greed colorblindness.

[129-131]: This is an interesting observation. The coupling networks of isobutyronitrile and butyronitrile are very different (A2B2C3 vs. AB6; A = H-1, B = H-2, C = H-3), so is the flow of polarization to the protons further away from the nitrile enhanced or suppressed by the type of spin network? Or are the couplings simply much smaller to the H-2 protons in isobutyronitrile?

[172/173]: “Therefore, we can expect more…” This sentence is incomplete.

Author Response

We are very grateful for your efforts to improve quality of our study. We willingly accept and answer your comments of critical issue on this study and revised our manuscript as you advised. Please see the attachment.

Reviewer 2 Report

In this manuscript, Kim and co-workers describe a systematic study of nitrile compounds using SABRE hyperpolarization technique. This work maybe publishable subject to addressing of the following concerns/criticism:

  • There is a great deal of emphasis of nitriles as biomolecular carriers of hyperpolarization for potential in vivo applications. Indeed, as the authors correctly describe that nitrile moiety occurs in a wide range of drugs. This may be true, but most nitrile-containing drugs are typically given in a small dose: a few milligrams to a few tens of milligrams. How do the author envision an in vivo administration of clinically-suitable dose? For a reference, 13C-pyruvate, which is in clinical trials now is given in the dose of ~1g.
  • My second concern with respect to potential biological utility of nitrile compounds is the authors demonstrate hyperpolarization of proton sites. They are relatively short lived: few seconds of 1H T1. How the compounds will be deployed if the lifetime of hyperpolarization is so short?
  • The experiments described are performed using a catalyst in methanol. For biological applications, the catalyst needs to be removed and material needs to be injected in a media with small (ca. <10%) content of alcohol (if any). This point needs to be clearly described a relevant literature needs to be cited.
  • Since the dynamics of SABRE process and hyperpolarization efficiency is a concentration dependent (Barskiy, D. A.; Knecht, S.; Yurkovskaya, A. V.; Ivanov, K. L. Nucl. Mag. Res. Spectrosc. 2019, 114-115, 33), it would be much better to present the results in mM concentrations rather than micromoles of liquid substrates.
  • Minor comment: The authors should carefully proofread the manuscript to avoid grammar challenges: e.g. “The parahydrogen generator was activated by a home-built instrument” should be replaced by “The home-built parahydrogen generator was employed to production of parahydrogen gas with ~50% para- enrichment” or something like that. There are many other little issues.
  • The authors claim that the reported here study is first in kind with respect to hyperpolarization of nitrile compounds. This is definitely not true. Hyperpolarization of three nitriles (including drug alectinib) have been reported previously in the following papers: Barskiy, D. A.; Shchepin, R. V.; Tanner, C. P. N.; Colell, J. F. P.; Goodson, B. M.; Theis, T.; Warren, W. S.; Chekmenev, E. Y. ChemPhysChem 2017, 18, 1493 and Colell, J. F. P.; Logan, A. W. J.; Zhou, Z.; Shchepin, R. V.; Barskiy, D. A.; Ortiz, G. X.; Wang, Q.; Malcolmson, S. J.; Chekmenev, E. Y.; Warren, W. S.; Theis, T. Phys. Chem. C 2017, 121, 6626. The level of polarization and signal enhancements obtained in those previous studies exceeded those reported here by an order of magnitude! These previous works need to be properly references and discussed.

Author Response

(The authors gave the same response as above.)

Round 2

Reviewer 2 Report

All comments have been addressed.